

# Development and validation of bioimpedance prediction equations for fat-free mass in unilateral male amputees

Hyuk-Jae Choi[1,*], Chang-Yong Ko[2,*], Yunhee Chang[1], Gyoo-Suk Kim[1], Kyungsik Choi[3] and Chul-Hyun Kim[4]

[1] Department of Medical Convergence Research & Development, Rehabilitation Engineering Research Institute, Incheon, Republic of Korea
[2] Department of Research & Development, Refind Inc, Wonju, Gangwon-do, Republic of Korea
[3] Department of Healthcare Business Division, Healthmax company, Seoul, Gangnam-gu, Republic of Korea
[4] Department of Sports Medicine, Soonchunhyang University, Asan, Chungcheongnam-do, Republic of Korea
[*] These authors contributed equally to this work.

Corresponding author
Chul-Hyun Kim, kimch37@sch.ac.kr

## ABSTRACT

**Background**. Metabolic disease due to increased fat mass is observed in amputees (APTs), thereby restricting their activity. Systemic health management with periodic body composition (BC) testing is essential for healthy living. Bioelectrical impedance analysis (BIA) is a non-invasive and low-cost method to test BC; however, the APTs are classified as being exempted in the BIA.

**Objective**. To develop segmental estimated regression equations (sEREs) for determining the fat-free mass (FFM, kg) suitable for APTs and improve the accuracy and validity of the sERE.

**Methods**. Seventy-five male APTs participated in this cross-sectional study. Multiple regression analysis was performed to develop highly accurate sEREs of BIA based on independent variables derived from anthropometric measurements, dual-energy X-ray absorptiometry (DXA), and BIA parameters. The difference in validity between the predicted DXA and sum of the segmentally-predicted FFM values by sEREs (Sum_sEREs) values was evaluated using bivariate linear regression analysis and the Bland–Altman plot.

**Results**. The coefficient of determination ($R^2$) and total error ($TE$) between DXA and Sum_sEREs were 71% and 5.4 (kg) in the cross-validation analysis.

**Conclusions**. We confirmed the possibility of evaluating the FFM of APTs through the sEREs developed in this study. We also identified several independent variables that should be considered while developing such sEREs. Further studies are required to determine the validity of our sEREs and the most appropriate BIA frequencies for measuring FFM in APTs.

## INTRODUCTION

Amputation refers to the removal of the upper or lower extremities to resolve the cause of a disability due to disease or trauma (*Hecht, MedicineNet.com & Shiel, 2003*). Physical and functional constraints are more severe in amputees (APTs) than in the general population (*Coffey, Gallagher & Desmond, 2014*). Previously, most amputations were performed following war-related or occupational injuries. Indeed, an increase in the number of APTs was observed following World War II and the Iraq War (*Kulkarni et al., 1998*; *Robbins et al., 2009*).

In addition, cardiovascular diseases such as high blood pressure/stroke and metabolic diseases including diabetes mellitus may necessitate amputation (*Kopelman, 2000*; *Rahman & Berenson, 2010*). Currently, reports suggest that the rate of amputation continues to increase worldwide. Notably, the number of APTs in the United States is predicted to increase from approximately 1.6 million (2008 data) to 3.6 million by 2050 (*Ziegler-Graham et al., 2008*).

Diabetes is associated with impaired blood supply to the extremities, which may lead to necrosis or severe ulcers that necessitate amputation (*Mishra et al., 2017*). Post-amputation complications, such as atrophy, phantom limb pain, and contracture, may lead to further activity constraints beyond those imposed by the amputation, thereby exerting deleterious effects on one's overall health (*Gallagher, Allen & Maclachlan, 2001*; *O'Sullivan et al., 2019*; *Ustun et al., 2003*). Additional post-amputation complications include obesity or excessive fat accumulation (*Kurdibaylo, 1996*), atrophy due to decreased muscle mobilization for ambulation and joint stability, and decreased physical strength (*Centomo et al., 2008*; *Isakov et al., 1996*; *Renstrom, Grimby & Larsson, 1983a*; *Renstrom et al., 1983b*; *Sadeghi, Allard & Duhaime, 2001*; *Bukowski, 2006*; *Zachariah et al., 2004*).

These complications may lead to an overall deterioration in physical health, which can be reflected by changes in BC and muscle condition, ultimately resulting in a decreased quality of life (*Chin et al., 2002*; *Rosenberg et al., 2013*; *Suk, Bom & Do, 2001*). Periodic assessments of BC and weight management interventions are critical for preventing secondary complications of amputation, such as excessive body fat/obesity (*Stone et al., 2006*; *Yoo, 2014*). Such assessments and interventions may not only help prevent obesity and muscle atrophy, but may also improve overall physical/mental health and quality of life.

Dual-energy X-ray absorptiometry (DXA) allows the measurement of bone mineral content (BMC), fat mass (FM), and soft lean tissue mass (SLTM) by passing two X-ray beams through the body. Although DXA is a highly accurate, validated method for the assessment of fat-free mass (FFM) (i.e., BMC+SLTM), these assessments are time-consuming and expensive. Furthermore, DXA requires exposure to small amounts of radiation, and some participants may be uncomfortable given the limited measurement space. Despite these limitations, DXA continues to be widely touted as the gold standard for determining BC (*Janssen, Heymsfield & Ross, 2002*; *Lee & Gallagher, 2008*; *Woodrow, 2009*).

Commercially available bioimpedance devices include single- and multi-frequency impedance analysis (SFBIA and MFBIA) and bioimpedance spectroscopy (BIS) (*Mulasi et al., 2015*). The principle of bioelectrical impedance analysis (BIA), including SFBIA, MFBIA, and BIS methods is to determine electrical impedance as "resistance (*R*)" (*Kyle et al., 2004a*) for total body water (TBW) and "reactance (*Xc*)" for body cell mass (*Walter-Kroker et al., 2011*). BIA is advantageous in that it is faster, more useful, less invasive, less physically restrictive, and less expensive than DXA. In addition, BIA equipment occupies far less space and requires less effort to operate than DXA equipment with a non-portable nature, which limits its use (*Buckinx et al., 2015*; *Janssen, Heymsfield & Ross, 2002*). As a routine clinical BIA tool, BC analysis was now readily available in a wider range of clinics and in the community (*Ward, 2019*). Given these advantages, BIA is performed in various settings. To estimate BC, BIA devices measure TBW content by sending a micro-current of less than 800 µA throughout the body. An estimated regression equation (ERE) is then used to predict FM and FFM (kg) (*Kyle et al., 2004a*). Recently, BIA has been used for the quantitation of BC through a specific mathematical model without empirically derived variables for athletic players (*Sardinha et al., 2020*) (*Campa & Toselli, 2018*; *Toselli et al., 2020*), children (*Colica et al., 2018*), older adults (*Silveira et al., 2020*), and young patients with cystic fibrosis (*Charatsi et al., 2016*).

However, APTs are typically excluded from such BIA studies owing to differences in limb length between the amputated and sound sides and irregular limb shapes (*Dogan et al., 2012*). Since the heights of the two sides differ, it is against the impedance index (*ZI*) formula (height$^2$/impedance), which is a more significant single predictor of FFM than other anthropometric variables (*Nguyen et al., 2007*).

In general, BC is predicted using the wrist-ankle method and variables, which consider the whole-body height for reducing typical BIA errors. Thus, the use of BIA equations, which are based on individuals (*Beaudart et al., 2020*) without amputation may not be accurate and valid when measuring the BC of APTs.

Therefore, we intended to develop EREs of APT using the variable of ZI and considering the segmental length based on Tanaka's study (*Tanaka et al., 2007*). Appropriate segmental estimated regression equation (sERE) for BC in APTs should consider residual limb properties, such as length, various other factors as independent variables including age, height, weight, gender, *ZI*, *R*, *Xc*, and phage angle (*PA*) without segmental amputated limb weight that is not measurable.

In this study, the InBody S10 BIA instrument (InBody S10) was used. This instrument allows for segmental analysis of various cellular properties through frequencies. Adhere-type electrodes are placed at eight precise tactile points to perform BIA in a comfortable and safe supine position for ATPs.

In this study, DXA measurements were used as a reference standard to develop EREs for BIA of APTs. The development of a valid ERE for use in APTs may improve their health management by providing accurate and convenient assessments of FFM through BIA. We confirmed the possibility of evaluating FFM of APTs through the sERE developed in this study. For an overview of abbreviations and parameters, see Table 1.

**Table 1 Abbreviations and concepts.**

| Abbreviation | Description |
|---|---|
| General parameters: | |
| BC | Body composition |
| BIA | Bioimpedance analysis |
| BIS | Bioimpedance spectroscopy |
| BPL | Body part length |
| DXA | Dual-energy X-ray absorptiometry |
| ERE | Estimated regression equations |
| MFBIA | Multi-frequency bioimpedance analysis |
| ROI | Regions of interest |
| sEREs | Segmental estimated regression equations |
| SFBIA | Single-frequency bioimpedance analysis |
| Sum_sEREs | Sum of the segmentally-predicted FFM values by sEREs |
| Subjects parameters: | |
| APTs | Amputees |
| LA, RA, LL, RL, TR | Left arm (LA), Right arm (RA), Left leg (LL), Right leg (RL), Trunk (TR) |
| Physiological parameters: | |
| BMC | Bone mineral content (kg) |
| FFM | Fat-free mass (kg) |
| FM | Fat mass (kg) |
| SLTM | Soft lean tissue mass (kg) |
| TBW | Total body water (L) |
| BIA parameters: | |
| PA | Phage angle (°) |
| R | Resistance (Ohm, Ω) |
| Xc | Reactance (Ohm, Ω) |
| Z | Impedance (Ohm, Ω). |
| ZI | Impedance index ($ZI = Height^2/Z$) |
| Statistical analysis parameters: | |
| LOA | Limits of agreement |
| $R^2$ | Coefficient of determination |
| SEE | Standard error of estimate |
| TE | Total error |
| VIF | Variance inflation factor |

# MATERIALS & METHODS

## Participants

This study was approved by the Institutional Review Board (No. 1040875-201707-SB-030 and RERI-IRB-190924-1). After receiving a complete description of the study, all participants provided written informed consent. A total of 75 male, unilateral APTs were recruited and included in the study. The mean age of the participants was 43.6 ± 12 years. Seventeen participants had previously undergone upper limb amputation (trans-humeral

 

**Table 2  Participant characteristics.**

|  | Upper-limb APTs ($n = 17$) | | Lower-limb APTs ($n = 58$) | |
| --- | --- | --- | --- | --- |
|  | Trans-humeral ($n = 5$) | Trans-radial ($n = 12$) | Trans-femoral ($n = 32$) | Trans-tibial ($n = 26$) |
| Age (year) | $42.6 \pm 5.7^{*}$ | $50.4 \pm 11.6$ | $41.1 \pm 13.3$ | $43.6 \pm 10.5$ |
| Height (cm) | $174.6 \pm 4.2$ | $168.3 \pm 7.2$ | $172.0 \pm 5.9$ | $171.4 \pm 5.6$ |
| Weight (kg) | $78.7 \pm 4.0$ | $74.2 \pm 7.6$ | $73.7 \pm 13.7$ | $73.2 \pm 12.5$ |
| BMI (kg/m) | $25.9 \pm 2.3$ | $26.2 \pm 2.0$ | $24.8 \pm 3.8$ | $24.9 \pm 3.6$ |
| Residual limb Length (cm) | $21.4 \pm 3.1$ | $42.7 \pm 4.9$ | $34.6 \pm 6.2$ | $62.2 \pm 6.7$ |
| Onset (year) | $13.6 \pm 7.9$ | $16.8 \pm 11.8$ | $13.2 \pm 11.9$ | $18.2 \pm 13.9$ |

**Notes.**

*Mean ± SD; APTs, amputees; BMI, body mass index; SD, standard deviation.

amputation: $n = 5$; trans-radial amputation: $n = 12$), and 58 had previously undergone lower limb amputation (trans-femoral amputation: $n = 32$; trans-tibial amputation: $n = 26$). APTs with disarticulation and multilateral (bi-, tri-) amputation were excluded. Additional participant characteristics including residual limb length (cm) and onset (postoperative period, years) are presented in Table 2.

## Experimental device (DXA)

DXA (Lunar Corp., Madison, WI, USA) measurements of FFM were used as the reference standard in the development of the ERE. This instrument was calibrated through the spine phantom provided by the manufacture daily. To standardize the scan, files from the original DXA system were transferred to iDXA software, version 4.0.2. The scan process was blinded and fulfilled by one radiologist who wore protective clothing. The segmentation method based on *Heymsfield et al. (1990)* was applied to uniform measurement (*Heymsfield et al., 1990*).

Participants were instructed to wear comfortable clothing for the assessment. Under the guidance of a professional examiner, each participant was asked to lay comfortably in the supine position on the assessment table, and spread both the upper and lower limbs. The whole-body DXA was performed for approximately 15 min.

## Experimental device (BIA)

BIA of the tetrapolar 8-point electrode type (InBody S10 for the supine measure, InBody Co. South Korea) was used in this study. This BIA model uses eight electrodes positioned at each hand and foot and enables multifrequency impedance measurement of the arms, trunk, and legs. Impedance parameters were measured with alternating current of 80 and 100 mA at frequencies of 1, 5, 50, 250, 500, and 1,000 kHz for InBody S10. After checking for precision errors of FFM about repeatability through biplicate measurements of the same APT based on a previous study (*Buckinx et al., 2015*), the InBody S10 was used. It was designed for single measurements in the supine position on a non-conductive surface through a SFBIA of only 50 kHz. In the sound limb, defined anatomical sites were cleaned with alcohol, on which adhesive gel electrodes were placed on the dorsal surfaces of the hand, wrist, ankle, and foot as follows: the proximal edge of the wrist electrode was

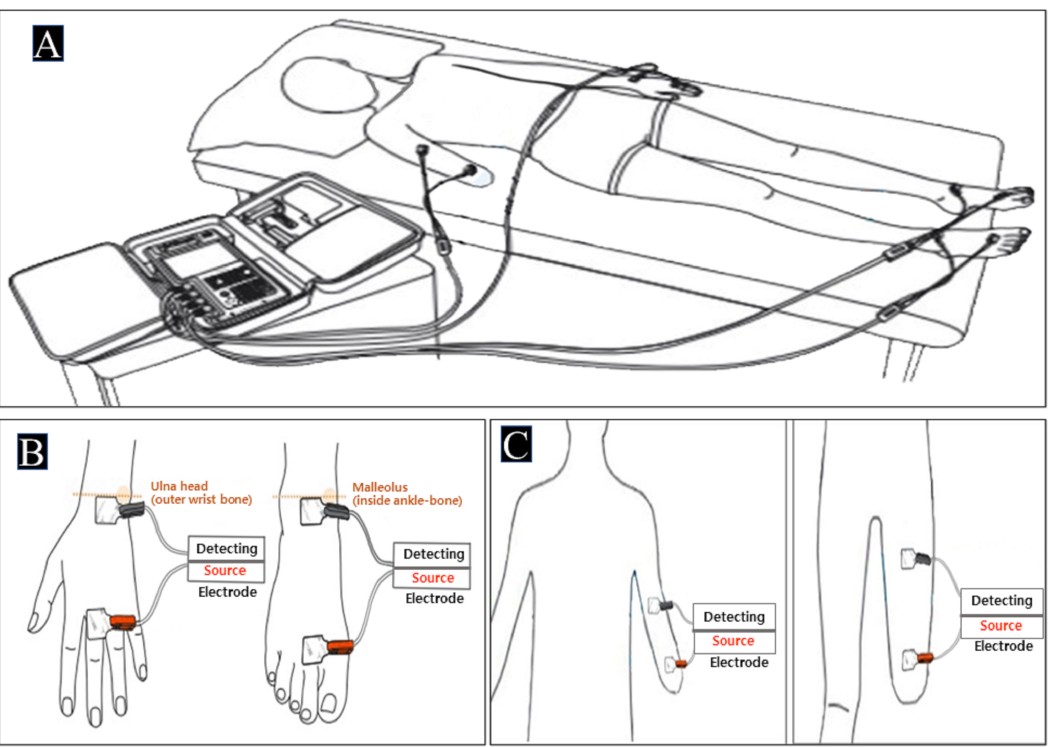

**Figure 1** **The testing postures and the electrode placements.** (A) InBody S10 in the supine position [permitted from the manufacture] (B) Electrode placements of sound limbs (C) Electrode placements of residual limbs.

attached from an imaginary line bisecting the styloid process of the ulna and the proximal edge of the finger electrode on an imaginary line from the imaginary line bisecting the metatarsophalangeal joint of the middle finger. The proximal edge of the ankle electrode was attached from an imaginary line bisecting the medial malleolus and the distal edge of the toe electrode was placed from an imaginary line through the metatarsophalangeal joints of the second toe as shown in Fig. 1B. In the residual limb, the distal and proximal electrodes were attached from the end of the stump (=distal part) to the region (=proximal part) by keeping the distance according to the instructions of InBody S10. Additionally, fixed-distance of electrodes was used with a 5-cm standard distance as shown in Fig. 1C (*Kaysen et al., 2005*; *Kriemler et al., 2009*). The device was calibrated every morning using the standard control circuit supplied by the manufacturer. We confirmed that the precision error was less than 2% (Fig. 1).

## Definition of segmental ZI (ZI) and regions of interest (ROI) in DXA

Impedance indices (*ZI*) for each body part were determined considering the residual limb length in each participant. *ZI* values were calculated in accordance with methods described by *Tanaka et al. (2007)*. As shown in Eq. (1), *ZI* was calculated by dividing height$^2$ by *Z* (based on values for non-APTs). We then calculated the body part length *ZI* ($ZI_{BPL}$) by dividing the body part length (BPL)$^2$ by *Z*. (*Tanaka et al., 2007*). BPL was measured for the

following five areas: left arm (LA), right arm (RA), left leg (LL), right leg (RL), and trunk (TR). The reference positions for the lengths of the upper/lower limbs and trunk were defined as follows: the area between the humeral head (=acromion) and styloid processes of the wrist (ulna) for the upper limbs; the area from the anterior superior iliac spine to the medial malleoli for the lower limbs; and the posterior length between the seventh cervical vertebra prominens to the center of the posterior superior iliac spine for the trunk (*Beattie et al., 1990*; *Hackenberg et al., 2003*; *Jamaluddin et al., 2011*; *Lee, Cha & Lee, 2016*; *Neelly, Wallmann & Backus, 2013*; *Ross, 1972*). The results of the experiment were analyzed by matching the physical measurements to the BIA electrode locations and DXA ROI based on these measurement standards.

According to the Heymsfield's protocol (1990), the boundaries of the ROI are defined as follows: (1) for the upper limbs of the ROI (right and left), the arms are isolated by running a line through the humeral head and (2) for the lower limbs, the pelvis cut is placed just above the pelvic brim and the computer automatically draws the lower pelvic lines to bisect the hip joints (*Heymsfield et al., 1990*; *Jeon et al., 2020*).

$$ZI = \frac{H^2}{Z} \tag{1}$$

H: Height of the whole body, Z: Impedance

$$ZI_{BPL} = \frac{BPL^2}{Z} \tag{2}$$

BPL: Body part length, Z: Impedance

**Independent variables for segmental BIA**

Independent variables included in the EREs were determined for the five body areas as follows: $R$ was applied by differentiating among the LA ($R_{LA}$), RA ($R_{RA}$), LL ($R_{LL}$), RL ($R_{RL}$), and trunk ($R_{TR}$). Using notations identical to those for $R$, impedance ($Z$), reactance ($Xc$), and PA were also calculated for each body part and expressed in terms of $R_{BPL}$, $Z_{BPL}$, $Xc_{BPL}$, and $PA_{BPL}$.

**Experimental procedures**

Prior to the measurements of FFM, participants were instructed to abstain from excessive dehydration-accompanied exercises and excessive alcohol use. In addition, they were instructed to fast for at least 6 h and abstain from alcohol consumption for at least 4 h.

We ensured that the urinary bladder was voided in all participants within 30 min before measurement, all participants wore non-conductive and comfortable sportswear. All conductive materials, prosthetic limbs, and amputation covers (silicone, amputation protection, etc.) were removed. First, DXA and then BIA test was conducted. For the measurements, after lying on a DXA sheet, stability (stabilization of BC) measurement was taken for 5 min in the supine position considering the potential for variable fluid shifts, and the DXA scan was then performed, and BIA was measured on the place immediately after the DXA scan without changing the position. The shoulder and hip joints of all participants were maintained at an abduction angle of approximately 15° for the shoulder and hip joints.

The elbow and knee joints were extended in a straight anatomical position. The physical contact of each electrode was ensured in accordance with the criteria recommended by the manufacturer. Each measurement took approximately 5–15 min. Participants were instructed to maintain a comfortable position without any movement during the examination (*Brantlov et al., 2017*; *Kyle et al., 2004d*).

**Sum of the segmentally-predicted FFM values by sERE (Sum_sEREs)**
After the development of the sEREs of the five body parts in the APTs, the sum of the segmentally-predicted FFM values by sERE was calculated. We confirmed the validity and accuracy between DXA and the sum of sERE_BIA about FFM through bivariate linear regression and the Bland_Altman plot.

**Statistical analysis**
The physical characteristics of the APTs group are presented as means with SDs. A normality test is required if there are fewer than 30 participants. However, since the number of participants in our study was 75, we assumed normality and analyzed all data (*Kwak & Kim, 2017*; *Sang Gyu et al., 2019*). The forward stepwise multiple linear regression analysis was used to develop sEREs in the APTs group. The significance level was set to $p < 0.05$. Variables included in the initial analyses contained $ZI_{BPL}$, $R_{BPL}$, $Xc_{BPL}$, $PA_{BPL}$, age (yr), height (cm), and weight (kg). The developmental equations were selected by measures of goodness-of-fit statistics, including coefficient of determination ($R^2$), the standard error of estimate (SEE), acceptable subjective rating of SEE (i.e., good to excellent) according to the minimally acceptable standard for prediction errors (*Buckinx et al., 2018*; *Heyward & Wagner, 2004a*; *Heyward & Wagner, 2004b*; *Lohman, 1992*), and the variance inflation factor (VIF). The SEE measures the variation in the actual values from the predicted values. The SEE represents the degree of deviation of individual scores form the regression line. It is computed using the following formula:

$$\text{SEE} = \sqrt{\sum(Measured\ \text{FFM} - Estimated\ \text{FFM})^2/(N-p-1)}$$

where $p$ = number of predicter variables. The VIF assesses how much the variance of an estimated regression coefficient increases when predictors are correlated for estimating collinearity/multicollinearity. In case of values more than 10, it can be assumed that the regression coefficients are poorly estimated due to multi-collinearity to remove predictors from the model. In our study, with values less than 10 (*Lee et al., 2018*; *Wickramasinghe et al., 2008*), we could proceed with our regression analysis. In the cross-validation, the group predictive accuracy of the Sum_sEREs was tested by calculating $R^2$, total error (*TE*: The TE represents the degree of deviation from the line of identity using the formula:

$$\text{Total Error} = \sqrt{\sum(Measured\ \text{FFM} - Estimated\ \text{FFM})^2/N}),$$

and acceptable subjective rating of TE (*Heyward & Wagner, 2004a*; *Heyward & Wagner, 2004b*; *Lohman, 1992*). The individual predictive accuracy of these equations was also tested by Bland-Altman plots, whitch included the bias of the mean difference between measured values of DXA and predicted values of Sum_sEREs. We used the 95% limits of agreement (LOA) between equations, and concordance correlation efficient ($r_{y-y',mean}$). Data were analyzed using Microsoft Office Excel Ver. 2013 (Microsoft, Redmond, WA, USA) and SPSS version 18.0 (IBM, USA).

**Table 3** The Final segmental estimated regression equations for FFM (kg).

| | |
|---|---|
| $LA_{FFM}$ | $y = -3.759 + 0.204(ZI_{LA}) + 0.410(Xc_{LA}) + 0.019(height) - 0.007(age)$ |
| | $R = .948 \qquad R^2 = .898 \qquad$ Adj. $R^2 = .892 \qquad$ SEE $= 0.286$ |
| | VIF: $ZI_{LA} = 1.286$, $Xc_{LA} = 1.193$, height $= 1.378$, age $= 1.274$ |
| $RA_{FFM}$ | $y = -1.370 + 0.212(ZI_{RA}) + 0.054(Xc_{RA})$ |
| | $R = .858 \qquad R^2 = .736 \qquad$ Adj. $R^2 = .729 \qquad SEE = 0.402$ |
| | VIF: $ZI_{RA} = 1.002$, $Xc_{RA} = 1.002$ |
| $LL_{FFM}$ | $y = -4.089 + 0.162(Xc_{LL}) + 0.143(ZI_{LL}) + 0.039(weight) + 0.006(R_{LL})$ |
| | $R = .953 \qquad R^2 = .908 \qquad$ Adj. $R^2 = .902 \qquad$ SEE $= 0.909$ |
| | VIF: $Xc_{LL} = 4.712$, $ZI_{LL} = 1.593$, weight $= 1.451$, $R_{LL} = 4.465$ |
| $RL_{FFM}$ | $y = -3.715 + 0.009(R_{RL}) + 0.152(ZI_{RL}) + 0.139(Xc_{RL}) + 0.031(weight)$ |
| | $R = .968 \qquad R^2 = .937 \qquad$ Adj. $R^2 = .933 \qquad$ SEE $= 0.739$ |
| | VIF: $R_{RL} = 8.942$, $ZI_{RL} = 1.460$, $Xc_{RL} = 8.415$, weight $= 1.166$ |
| $TR_{FFM}$ | $y = -12.061 + 0.046(ZI_{TR}) + 0.073(weight) + 0.212(height) - 0.419(R_{TR}) + 0.041(age)$ |
| | $R = .880 \qquad R^2 = .775 \qquad$ Adj. $R^2 = .758 \qquad$ SEE $= 1.510$ |
| | VIF: $ZI_{TR} = 2.441$, weight $= 1.649$, height $= 1.823$, $R_{TR} = 1.709$, age $= 1.294$ |

**Notes.**

FFM, fat-free mass (kg); $ZI_{BPL}$, body part length impedance index; $Z_{BPL}$, impedance; $Xc_{BPL}$, reactance; $R_{BPL}$, resistance; LA, left arm; RA, right arm; TR, trunk; LL, left leg; RL, right leg; Adj., Adjusted; VIF, variation inflation factor; SEE, standard error estimate (kg).

## RESULTS

### Segmental Estimated Regression Equation

DXA measurements of FFM were used as the dependent variable in the development of the EREs for use in APTs. Various independent variables were entered to ensure optimal model development. Our model considered factors such as $R^2$, multicollinearity (tolerance and variance inflation factor [VIF]), and standard error estimates (SEE). Using these factors, we developed sEREs for the left and right upper/lower limbs as well as the trunk. $ZI_{LA}$, $Xc_{LA}$, height, and age were entered as independent variables in the final sERE model for FFM in the left arm ($LA_{FFM}$). Values for the final sERE for $LA_{FFM}$ were as follows: $R = 0.95$, $R^2 = 0.90$, and adjusted $R^2 = 0.89$. $ZI_{RA}$ and $Xc_{RA}$ were entered as independent variables in the final sERE model for FFM in the right arm.

Values for the final sERE for RA $_{FFM}$ were as follows: $R = 0.86$, $R^2 = 0.74$, and adjusted $R^2 = 0.73$. $R_{BPL}$, $Xc_{BPL}$, $ZI_{BPL}$, and weight were entered as independent variables in the final models for both the left and right lower limbs. The $LL_{FFM}$ model included $R_{LL}$, $Xc_{LL}$, $ZI_{LL}$, and weight. The final sERE values for $LL_{FFM}$ were as follows: $R = 0.95$, $R^2 = 0.91$, and adjusted $R^2 = 0.90$. The highest correlation coefficients were observed for $RL_{FFM}$: $R = 0.97$, $R^2 = 0.94$, and adjusted $R^2 = 0.93$. In contrast, the lowest correlation coefficients were observed for $TR_{FFM}$: $R = 0.88$, $R^2 = 0.78$, and adjusted $R^2 = 0.76$. A total of five independent variables were entered for the $TR_{FFM}$ ERE: $ZI_{TR}$, weight, height, age, and $R_{TR}$ (Table 3).

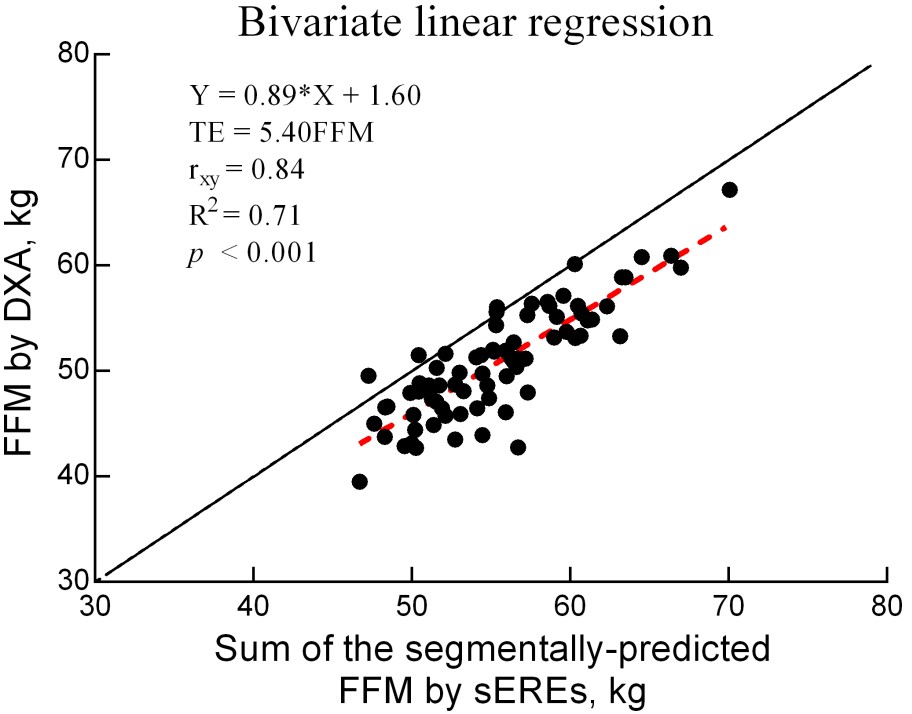

**Figure 2  Bivariate linear regression for FFM values obtained using DXA and sEREs.** FFM: fat-free mass (kg), sEREs: segmental estimated regression equations, TE: total error, r: validity coefficient, DXA: dual-energy X-ray absorptiometry.

## Cross-validation between BIA and DXA about FFM
### Linear regression, total error, and line of identity

Linear regression analyses were used to calculate the correlation between BIA estimates based on the final segmental ERE and standard DXA measurements. In the total error (*TE*) calculation (Eq. (3)), Y–Y' represents the difference between the DXA measurement (Y) and the BIA estimate (Y'), while N represents the sample size.

$$TE = \sqrt{\sum (Y - Y')^2 / N} \tag{3}$$

The *TE* for the comparison between DXA and BIA values for FFM was 5.4 (kg), with a correlation of 0.84 ($p < 0.05$) (Fig. 1).

## Residuals and bias in Bland–Altman plots

To evaluate the validity of the final ERE, the residuals (i.e., the difference between BIA-estimated and DXA-measured FFM) and means of the two methods were assessed using the Bland–Altman plot (Fig. 2). The bias of the difference between the two FFM measurements was −4.60 kg. BIA estimates obtained using the ERE tended to be higher than those obtained using DXA values. Furthermore, there was a tendency for the residuals of the bias to be evenly distributed, and increased bias tended to be associated with decreased residuals.

## DISCUSSION

In the present study, we aimed to develop sERE for BIA in APTs using DXA measurements as the reference standard. Given the correlation ($r = 0.85$), $TE$ (5.4 kg), and coefficient of determination ($R^2 = 71\%$) between FFM values obtained using Sum_sEREs and DXA, our findings confirmed that the sERE developed in this study could assess FFM in APTs.

Based on findings obtained in previous studies, sex differences should be considered in the selection of independent variables to obtain more accurate estimates (*Beaudart et al., 2020*). However, this study was conducted on only male APTs because of a difference in the sex ratio within the total APTs recruited; hence, representativeness of sex could not be achieved. Furthermore, data from the Multidimensional Body–Self Relations Questionnaire suggest that women experience significantly higher level of dissatisfaction with their bodies than do men following amputation (*Holzer et al., 2014*). For the above reasons, our sEREs were developed using data from male APTs only.

The process of developing sERE was carried out by applying the FFM of DXA on the same limb BIA variables. Therefore, the dominant and non-dominant limbs were not considered specifically. Additionally, in the process of validation of the developed sERE for each limb, the same limb variables were applied. For example, the DXA value of the right leg of the amputated side limb and the BIA of the same right limb were compared.

There are examples for various types of amputations for uni-, bi-, and multi-lateral APTs. However, in this study, for the development of a basic sERE, only unilateral APTs were included to control the length variable of the amputated site. Moreover, given the difficulty in controlling for the BPL among multilateral APTs, we restricted our participants to unilateral APTs.

The measurement range of the InBody S10 device (Biospace, Korea) extends from 5 to 1,000 kHz (1, 5, 50, 250, 500, and 1,000 kHz). MFBIA can more accurately measure intra- and extracellular body water than SFBIA of 50 kHz. However, SFBIA of 50 kHz is better for measuring cell membrane properties through Xc, because it provides equivalence of information for the function of Xc at 50 kHz versus other frequencies (*Piccoli et al., 2005*). For developing basic sERE of APTs, we carefully analyzed the cell membrane state of the APT rather than analyzing between the intra- and extracellular body water properties (*Kyle et al., 2004b*; *Heymsfield et al., 2005*). Only an SFBIA of 50 kHz was used for the calculation of total body water, on which estimations for FFM are based using proprietary equations (*Achamrah et al., 2018*).

Previous BIA studies have excluded APTs as well as patients with joint deformation, hemi-paralysis, and uncommonly large/small bodies (*Dogan et al., 2012*; *Kyle et al., 2004e*; *Mialich, Sicchieri & Junior, 2014*). *Tanaka et al. (2007)* proposed using a tetrapolar 4-point BIA to measure BC in such individuals (*Mialich, Sicchieri & Junior, 2014*; *Tanaka et al., 2007*).

However, in a 4-point BIA with the tetrapolar system, the electrodes are attached manually by connecting electrodes to measure the segments according to manufacturer's instructions. Four-point estimates are derived based on the characteristics of the flow of current on one side of the body, allowing for the calculation of BC for the specific body

part, following which total estimates are obtained. Thus, the measurements had substantial systematic errors, including underestimation or overestimation of accuracy (*Janssen et al., 2000*; *Tanaka et al., 2007*). Given the errors of the tetrapolar 4-point BIA, Foster and *Lukaski (1996)* highlighted the need for further research regarding the use of such measurements for the analyses of BC (*Lukaski, 1996*). To avoid these issues, we utilized the tetrapolar 8-point segmental BIA (sBIA) as InBody S10, in which the pairs of electrodes are attached to measure the different body segments.

Before 1980, the ERE for FFM only included the resistance index (height$^2$/resistance). *ZI* was calculated by taking into account the variables for the whole-body height in ERE of non-APTs. However, the sERE of APT developed in this study was analyzed by applying *ZI*$_{BPL}$ and BIA parameters considering the length of each body part, including the residual limbs and condition of this amputated region.

In five sEREs for each part, the redundancy of variables was considered through VIF and the highly accurate sEREs calculated through a meticulous analysis process based on the BIA characteristics that affect the APT.

The BIA variables applied to the sEREs are *R, Xc*, and *Z* as shown in Table 3. we were to confirm that a complex quantity composed of ($R$) which is caused by TBW, and capacitance of the cell membrane related to ($Xc$), and obstruction to the flow of an alternating current ($Z$) that was dependent on the frequency of the applied current based on the theoretical basis of BIA in the limbs of APT (*Kyle et al., 2004a*; *Kyle et al., 2004c*) (*Khalil, Mohktar & Ibrahim, 2014*).

Generally, assuming our body as a cylinder, both arms and legs are attached to the body, and the whole-body height is used for the *ZI* ($ZI$ =Height$^2$/Z). However, in this APT study, the length of each limb was substituted for the *ZI* of each limb ($ZI$ $_{BPL}$=BPL$^2$/Z) to make the estimation equation of the segmental limb considering amputated extremity. For example, in the process of developing estimating equations, such as the FFM of each of the right upper and left upper limbs etc., the final model with a low statistical error and a high estimating power was selected using the length and BIA variables of the same body part. The APTs had imbalance between the left and right limbs, similar to that reported in a previous study (*Sherk, Bemben & Bemben, 2010*), showing muscle and fat imbalance between the injured and sound limb; consequently, there was a difference in variables entered in the final sERE obtained from the final model selected as the criterion of the significance level ($P$<0.05), VIF (<10), SEE, and $R^2$.

In the present study, we utilized a forward stepwise multiple regression analysis that included diverse independent variables such as residual limb length. When developing an ERE, approximately 20 participants are required for each independent variable entered. Given that our study included 75 participants, four or fewer independent variables are considered appropriate (*Heyward & Wagner, 2004a*; *Heyward & Wagner, 2004b*). These independent variables included ZI$_{BPL}$, height, weight, age, onset, and segmental BIA factors.

Table 3 shows the final EREs. The number of variables entered in each sERE ranged from two to five. Although previous studies have specified that only four variables should be included, based on our sample size, it was necessary to consider factors, such as

$R^2$, multicollinearity (tolerance and VIF), and SEE, to develop the most ideal model. Nonetheless, the sERE for the trunk was the only equation to have been developed using five variables.

BIA estimates of FFM are based on TBW measurements, which are derived from $Z$ values obtained by passing microcurrents throughout the human body. In this calculation, the human body is assumed to be cylindrical (TBW $= \rho$ x height$^2$/Z, $\rho$=constant). Based on the theory that TBW comprises 73% of FFM (TBW $=$ FFM $\times$ 0.73), the estimates of FFM can be obtained using the following equation: FFM $=$ TBW/0.73. Factors, such as race, age, sex, and medical history, influence the unique conduction constant ($\rho$) as well as the correlation between TBW and $ZI$, making it necessary to include several independent variables in the TBW calculation (*Kyle et al., 2004b*). Therefore, in this study, we included additional independent variables, such as onset and characteristics of the body part amputated.

Several BIA studies conducted outside of Korea have included patients with hemi-paralysis (*Kafri, Potter & Myint, 2014*; *Nalepa et al., 2019*; *Yoo et al., 2016*), pediatric scoliosis (*Matusik, Durmala & Matusik, 2016*), or Turner's syndrome (i.e., abnormally small body)(*Guedes et al., 2010*). However, no such studies have been conducted on the APTs. Despite this, studies have recommended that BIA measurements be obtained in the non-amputated limb (*Kyle et al., 2004e*; *Mialich, Sicchieri & Junior, 2014*).

However, to estimate the whole-body FFM, the amputated body parts must be considered. In our study, the sEREs for FFM did not exhibit a close relationship with $PA_{BPL}$, and we did not include PA as an independent variable for any ERE in Table 3, similar to models developed using data from the general population (*Mialich, Sicchieri & Junior, 2014*). Meanwhile, $Xc$ has been utilized in numerous EREs for FFM in studies conducted outside Korea (*Lieberman, 1993*; *Kyle et al., 2001*; *Roubenoff et al., 1997*; *Stolarczyk et al., 1994*).

In accordance with previous findings, our sERE for FFM exhibited a close relationship with $Xc$, whereas onset did not appear to exert a significant impact on FFM estimates. Although changes in BC occur over time following amputation, FFM can be maintained with systematic rehabilitation and BC management during the first postoperative year. This was noted by a previous study of those for whom 2 to 15 years had passed since amputation (*Eckard et al., 2015*; *Renstrom et al., 1983b*).

In our study, the final sEREs were selected by considering factors, such as $R^2$, tolerance, VIF, and SEE. As shown in Fig. 2, the correlation for FFM values was high ($R = 0.84$), with an $R^2$ of 71%. This correlation is higher than the standard of 0.80 ($R^2 = 64\%$) suggested by Heyward and Wagner (*Heyward & Wagner, 2004a*; *Heyward & Wagner, 2004b*) for validity research. A bivariate linear regression equation (Y $=$ aX $\pm$ b) was used to confirm the accuracy of Sum_sEREs as well as the correlation. The slope (a) and y-intercept (b) were used to analyze the correlation between DXA and BIA measurements. The slope of the bivariate linear regression equation was 0.89, whereas the y-intercept was 1.60, yielding a simply positive correlation of y $=$ x. However, the slope of 0.89 exhibited a positive correlation that was close to the standard of 1. Furthermore, the y-intercept was close to

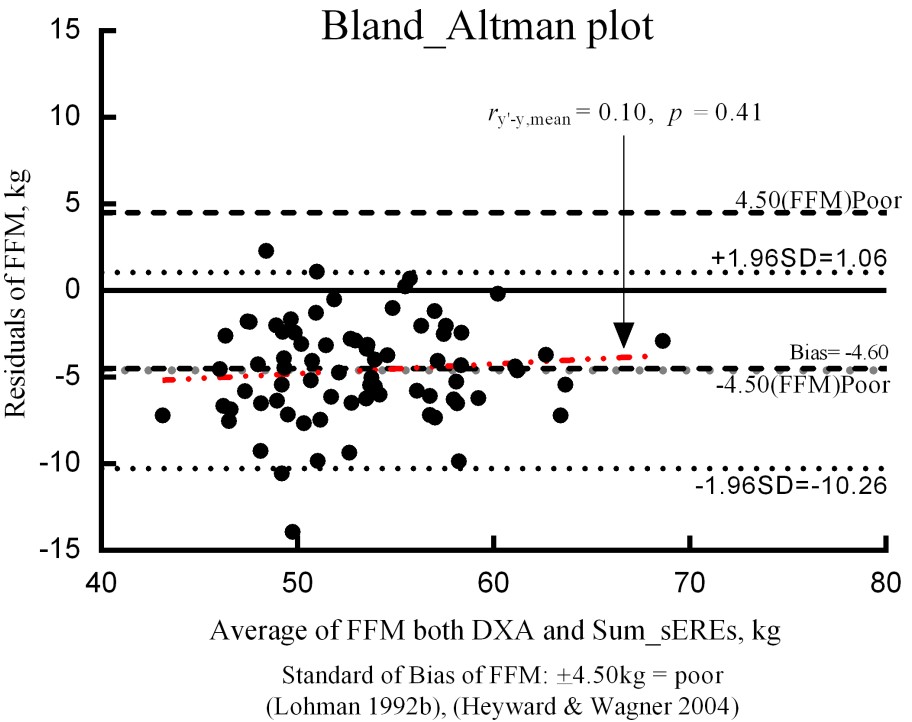

Standard of Bias of FFM: ±4.50kg = poor
(Lohman 1992b), (Heyward & Wagner 2004)

**Figure 3 Bland–Altman plot.** Bias: mean of DXA-BIA value, ± 4.5(FFM) poor = "poor" standard for evaluating prediction errors, FFM: fat-free mass (kg), DXA: dual-energy X-ray absorptiometry, sEREs: segmental estimated regression equations.

the ideal standard of 0. Taken together, these results suggest the possibility of sERE for APT.

Individual errors for DXA and BIA results are shown in the Bland–Altman plot in Fig. 3. The average value representing the difference between the two methods (i.e., Bias) was -4.60 kg. When the bias approaches 0 ( $y = 0$ ), there is no mean difference in the measurement values with an ideal validity. However, bias did not approach 0 in our study. BIA tended to overestimate FFM, relative to the value obtained using DXA. *Ainsworth et al. (1997)* suggested that the results are ideal if the proportion that exceeds the standard of overestimation and underestimation is less than 30% (*Ainsworth et al., 1997*). In our study, the proportion of APTs who exceeded the standard of bias was 50.7%, suggesting a need for further validity studies.

## Limitation

In this study, limitations were placed on the use of only 50kHz frequency to characterize the cell membrane; however, ICF and ECF have limitations in reflecting the characteristics at a normal level. We intend to proceed with the MFBIA study including 1, 50, 250, 500, 1000kHz, and over frequencies, as a future research project. In the validation procedure, we recognized the importance of validating a method through a split group design, K-fold or LOOV-type. However, the sEREs were developed without external cross-validation through group of predictive power test. We confirmed only cross-validation between DXA

and Sum_sEREs of FFM values. Additionally, we also did not perform a thoughtful analysis of the PA, but we intend to do it for the APTs in the future.

## CONCLUSIONS

In the present study, we utilized multiple regression analysis to develop sEREs for FFM in APTs, using DXA as the reference standard. Although there was a bias of $-4.598$, and LOA of $-0.26$ $1.06$ in our results, we could confirm the minimal clinical feasibility based on the coefficient of determination ($R^2 = 71\%$), and $TE$ (5.40 kg). In addition, we identified several independent variables that should be considered while developing such sEREs for APT. Further studies are required to determine the validity of our sEREs and the most appropriate BIA frequencies for measuring FFM in APTs.

### Funding
This study is part of the ICT innovative company technology development support project supported by the Ministry of Science and ICT in 2020 (2019-0-01758). The funders had no role in study design, data collection and analysis, decision to publish, or preparation of the manuscript.

### Grant Disclosures
The following grant information was disclosed by the authors:
Ministry of Science and ICT in 2020:  2019-0-01758.

### Competing Interests
Chang-Yong Ko is employed by Department of Research & Development, Refind Inc. and Kyungsik Choi is employed by Healthmax company. The authors declare there are no competing interests.

### Author Contributions
- Hyuk-Jae Choi performed the experiments, analyzed the data, authored or reviewed drafts of the paper, and approved the final draft.
- Chang-Yong Ko and Yunhee Chang performed the experiments, analyzed the data, prepared figures and/or tables, authored or reviewed drafts of the paper, and approved the final draft.
- Gyoo-Suk Kim analyzed the data, prepared figures and/or tables, authored or reviewed drafts of the paper, and approved the final draft.
- Kyungsik Choi analyzed the data, authored or reviewed drafts of the paper, and approved the final draft.
- Chul-Hyun Kim conceived and designed the experiments, authored or reviewed drafts of the paper, and approved the final draft.

## Human Ethics

The following information was supplied relating to ethical approvals (i.e., approving body and any reference numbers):

The University of Soonchunhyang and Rehabilitation Engineering Research Institute granted Ethical approval to carry out the study within its facilities (No. 1040875-201707-SB-030 and RERI-IRB-190924-1).

## Clinical Trial Ethics

The following information was supplied relating to ethical approvals (i.e., approving body and any reference numbers):

The University of Soonchunhyang granted Ethical approval to carry out the study within its facilities.

## Data Availability

Raw data are available as a Supplementary File.

## Clinical Trial Registration

The following information was supplied regarding Clinical Trial registration:

No. 1040875-201707-SB-030 and RERI-IRB-190924-1.

## Supplemental Information

Supplemental information for this article can be found online at http://dx.doi.org/10.7717/peerj.10970#supplemental-information.

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
