# Peer review of "Development and validation of bioimpedance prediction equations for fat-free mass in unilateral male amputees"

_PeerJ, doi:10.7717/peerj.10970_

## Round 0.1 · original submission · Major Revisions

The reviewers noted -- and I agree -- that the article addresses an interesting topic, but they both also note several areas where the article could be improved. I encourage the authors to consider the thoughtful reviewers' comments, and to submit a revised version that addresses them.

·

Basic reporting

A generally well-written manuscript with acceptable English although with flaws as noted below. Cited references are mostly recent, although not always the most recent, and provide adequate background to the study. Raw data have been provided. The manuscript is appropriately structured.

Experimental design

The experimental question is clearly stated and background provided. The experimental design is acceptable and execution allowed required raw data to be obtained. Unfortunately, the manuscript lacks necessary methodological detail to allow full replication and the data analysis could be improved.See below for details

Validity of the findings

The authors have not described adequately, exactly how the FFM has been calculated. The multiple regressions appear to have been undertaken simply to maximize the correlation and minimize error without thought as to the appropriateness of inclusion of variables. For example, it is hard to see why different predictor variables are required for the left versus the right arm.The conclusions are not justifiable. The authors appear to equate a strong correlation with predictive performance. The clinical value of the data and the clinical impact of errors are not discussed. These points are amplified in the section below.

Additional comments

Line 78 onward. BIA is a generic term used to describe the overall methodology. Within this there are single frequency BIA (SFBIA), multiple frequency (MFBIA) and spectrometric approaches (bioelectrical impedance spectroscopy, BIS). Each of these can be used for whole-body or segmental measurements and are not always exclusive, for example BIS can provide SFBIA data but not vice versa. Methods of transformation of measured impedance/resistance to body composition does not always use empirically-defined prediction equations. Hence this explanation of BIA is simplistic. The authors should rewrite this section to reflect the above comments and provide suitable citations to recent reviews. They may wish to highlight the approach to be used in the present study. It is not clear from the text exactly what type of device the S10 is.
Line 84. This is vague. What "independent variables"? I assume that you are referring to , weight, sex etc. in addition to BIA parameters but this is not clear.
Lines 89/90. Again this vague and potentially misleading. A simple BIA prediction of body composition based on wrist-ankle measurement on the non-amputated side could predict perfectly well (within typical BIA-related errors) since stature is used not individual segment lengths. It would, however, be in error if the ERE included weight as a variable since presumably the ERE would have been generated on a population of non-amputees, i.e. including amputated limb weight. The point being made is that the text is simplistic.
Line 94. While ERE is defined sERE is not.
Line 102. Why were the 10 females dropped from the analysis? Please justify, line 222 states simply the sample was not large enough, on what basis? Were they all unilateral amputees, were any bilateral? On what side were amputations, dominant or non-dominant?. Humans are bilaterally asymmetric, having more FFM on their dominant side, particularly in the arms. For upper-limb amputees, work-induced hypertrophy of the remaining limb may be a confounder. Was this considered?
Line 111. The description of DXA is limited. For example, it is well recognized that inter-device, inter-software differences exist. provide more details. Were these single measurements? Was a soft-tissue calibration performed (not simply for BMC/BMD)?
Line 118. This again lacks detail. For example, is the S10 device a stand-on, hand-held or electrode type device? Line 125, suggests the latter. What exactly was the measurement protocol? Were the participants lying down, if so for how long before measurements? Were measurements in replicate or single measurements? Since it is unlikely that manufacturer's instructions covered amputees where were electrodes located in these cases? The authors state that the device can perform measurements at adjustable frequencies in the range 1-1000 kHz. While it is implied, but not explicitly stated was 50 kHz the frequency of measurement chosen? How was the system calibrated and what QC procedures were in place? More detail is required.
Segmental impedance measurements. The use of resistance/impedance quotients is supportable by theory and is standard practice. My question here is why consider Z, R, Xc and phase? The aim was to produce EREs for FFM. Impedance is based on measuring the opposition to current flow through the body, i.e. body water and, assuming either implicitly or explicitly, an hydration fraction for FFM. The appropriate measure that relates to water volume (FFM) is R. Xc and phase relate to cell membranes and Z is a composite or R and Xc (Z^2 = R^2 + Xc^2). So all three parameters are intimately related. Hence there is a high potential for multi-collinearity. Similarly, because of this relationship it is generally considered inappropriate simply to include multiple combinations of these parameters in regression equations (Table 2). It also appears from Table 2 that absolute values for R (or Xc or Z) as well as the length-related quotients are included. Why? The authors explain this (line 247 onward) by simply saying that since the early studies additional variables have been included to improve accuracy. This is not good scientific reasoning. Others may have included additional variables wrongly. There should be valid reasons for their inclusion. Basic impedance theory, provided in many reviews, demonstrates that the quotient should be used not simply the raw R. This section needs to be re-written to align with the theoretical basis of BIA.
Since it has not been made clear where electrodes were actually placed, it is equally not clear how these align with limb lengths. For example, if electrodes were placed along the limb e.g. at the wrist and acromion then this length should be used. If the method of equipotentials was adopted then the proximal virtual electrode position is unknown and hence a surrogate length measurement should be used. Line 143, implies that electrodes were located at the length measurement locations and these were then also used for DXA ROI but it is not made clear.
Line 154. Again more details are required. What was the order of measurements, DXA then BIA or vice-versa? Was it consistent? If not, the time interval for the subject (assumed to be supine) lying down prior to BIA would be variable, and hence the potential for variable fluid shifts. More details needed.
Line 162. Statistical analysis is not well described. Multiple regression was used. Was normality assessed? What criteria for multi-collineraity was used - VIF threshold? Were potential outliers evaluated and if present what strategy was used to account for this. Step-wise can be forward or backward, what criteria were used to remove variables from the regression, indeed what variables were included and assessed? Fig 1 is a regression/correlation of predicted versus measured. What type of regression was used? Where both variables are subject to error then a procedure such as Deming or Passing-Bablok regression should be used. Correlation should be assessed using Lin's concordance correlation not Pearson. Some details are provided but in Results (line 191 onward). This should be in statistical analysis. The B&A plots are not a representative validation. All data were used to generate the ERE and then again used in the B&A plot. This is circular and will not provide a true measure of validity. Validation should either be via a split group design, K-fold or LOOV-type procedure. For example, an ERE in say 50 randomly-selected participants could be generated and the its predictive power tested in the remaining 25. There is ample published literature on these methods, including specifically for validating impedance predictors.
Results.
Table 2 and text. It is not made clear how the amputations are distributed between the different regression equations. For example, for upper limb amputees were they all the same side? Is a the distribution a contributor to why quite different equations (variables and coefficients) have been generated for the right and left arms? What is the reason that might explain height and age being significant predictor variables for the left arm but not the right? This would not seem logically plausible for non-amputees and would appear unlikely for this group also unless amputation was much more prevalent on one side than the other. Explanation is required.
Figs. Again it is not explicitly explained how these data are obtained. Is it a result of a whole-body wrist-ankle prediction or is it the sum of the segmentally-predicted values. Fig 2. On what criteria is a bias of 4.5 kg considered poor? What is clinically acceptable (minimal clinically important difference, MCID)?
Discussion
Line 212. The authors appear here to be confusing correlation with predictive performance. A high correlation does not necessarily indicate a good predictor in terms of accuracy. The authors have not actually assessed this for the segments only whole body. Bias was 4.5 kg for a mean FFM of around 50-52 kg. This is a bias of around 10% is this acceptable? More importantly there is systematic bias, well demonstrated in Fig 1. Why? This needs to be discussed. The LOA are -10.26 to 1.064 or around +/-6.0 kg, again around +/- 11%. Note that this is based on a circular B&A (see above), a true cross-validation would likely produce worse results. Is this clinically acceptable? This is important since in a clinical setting accuracy in an individual is of paramount importance. Again this requires discussion and a reasonable correlation (line 310) should not be used as justification for adequacy of the method.
Line 228. 50 kHz is not a standard, it is a commonly used frequency largely for historic reasons and from the widespread use of SFBIA devices. A strong argument can be made that if a single frequency is to be used it should be a the characteristic frequency or a frequency higher than 50 kHz where current penetration of the cell membrane is greater.
Lines 234 onward. This is misleading. The authors imply that measurement systems are 4 or 8 point. This is incorrect. ALL measurements are tetrapolar, two current drive electrodes and two proximal voltage sense electrodes. The so-called 8-point systems use the same measurement approach but simply swap in turn pairs of electrodes to measure the different body segments. A typical "4-point" system can achieve exactly the same by manually connecting electrodes to measure the segments according to the equipotential principle. There are many publications attesting to this. The use of the "8-point" system as implemented in the In-Body device is not justification for the claim of highly accurate sERE (line 246).
Line 293. I do not see the basis for the claim of "perfect positive correlation of y=x". This requires justification.

·

Basic reporting

- This paper is relevant, nicely written and has a logical structure.
- Relevant use of references.
- Use the abbreviation BC for body composition throughout the paper.
- The abbreviation of phase angle (PhA) should be changed to PA since this is the accepted abbreviation in the bioimpedance literature.
- Decide how to report figures in the paper, with 2 or 3 decimals. In your paper, I would use 2 decimals for R-values and kg (e.g. -4.598 kg).
- Line 119 and 226: Write 1000 kHz instead of 1,000 kHz

Experimental design

- In general, the experimental design is well described.
- Line 118-128: However, the use of the BIA technique needs more attention when reporting in order to obtain valid measurements. E.g. was measurements made in triplace?, was the device calibrated prior to use?, how was skin prepared before placement of electrodes?, how was the subjects positioned during measurements? etc. The importance of standardised bioimpedance measurements have been proposed by following authors:

https://www.tandfonline.com/doi/abs/10.1080/03091902.2016.1209590?journalCode=ijmt20

https://www.tandfonline.com/doi/full/10.1080/03091902.2017.1333165

https://pubmed.ncbi.nlm.nih.gov/15556267/


- Line 162-: Was data normally distributed?
- Line 163: Please add a p for statistical significance, e.g. p ≤ 0.05.

Validity of the findings

- Relevant to study prediction equation for use in amputees.
- The number of subjects included seems relevant.
- The data analysis models (Linear regression and Bland-Altman) are relevant when developing prediction equations.

---

## Round 0.2 · Minor Revisions

The authors have made a number of changes to the manuscript, addressing many of the reviewers' comments. There are still some outstanding issues raised by Reviewer #2 that should be addressed before the article can be accepted for publication. The Editor would appreciate if you could revise the points raised by Reviewer #2.

·

Basic reporting

The authors have satisfactorily addressed the issues raised in my original review. In places the English is somewhat idiosyncratic but the meaning remains generally clear.

Experimental design

The authors have satisfactorily addressed the issues raised in my original review. Additional methodological detail has been provided.

Validity of the findings

The authors have satisfactorily addressed the issues raised in my original review. The authors have adopted revised/improved statistical analysis.

Additional comments

The authors have satisfactorily addressed the issues raised in my original review.

·

Basic reporting

- This paper is relevant, generally nicely written and has a logical structure.
- Relevant use of references.

Experimental design

- In general, the experimental design is well described.
- Line 111-119 “Participants”. Why is there no control group?
- Line 136-137 “Experimental device (BIA)”: “After checking reproducibility and the precision error of FFM…” If you have checked reproducibility, then only write this since reproducibility is a precision measurement. Precision is the overall term containing repeatability (same person) and reproducibility (different persons).
- Line 132-152 “Experimental device (BIA)”: Was measurements made in triplicate?
- Line 168: Change to Heymsfield and remove region of interest and only keep ROI.
- Line 248: Write FFM in the subheading, since the terms BIA and DXA are used.
- Line 284: “SFBIA is better for measuring cell membrane properties through Xc”. Better than what? Why? Please elaborate on that statement.
- Line 287: “Only an SFBIA of 50 kHz was used for the calculation of total body water. Why that? In line 283 it is stated that “MFBIA can more accurately measure intra- and extracellular body water than SFBIA of 50 kHz. From the intra- and extracellular body water the total body water can be calculated. Please explain how this is related to the knowledge and potential of MFBIA and BIS techniques?
- Line 381-382: “We used only 50 kHz to characterize the cell membrane. We will proceed with the MFBIA study as a future research project”. Now you write that you only used 50 kHz to characterize the cell membrane. The use of “only” indicates that you are not sure whether 50 kHz is “enough” for measuring the property of the cell membrane. Once again, in line 284 you wrote that “SFBIA is better for measuring cell membrane properties through Xc”. Please explain how all these statements are related. Also, what is the MFBIA study?
- Figure 1: The picture has a bad resolution. Try to find another one.

Validity of the findings

- Relevant to study prediction equation for use in amputees.
- The number of subjects included seems relevant.
- The data analysis models (Linear regression and Bland-Altman) are relevant when developing prediction equations.

Additional comments

- Due to the numerous abbreviations and parameters used in the paper, which makes it a little hard to read the paper, please add a Table summarizing these.
- The first time an abbreviation is used, it must be written in full. Be consistent with abbreviations throughout the paper.
- I find the use of phase angle (PA) a little awkward in the paper. Maybe it should be analysed separately in later paper?

---

## Round 0.3 · accepted · Accept

Thank you for your submission to PeerJ.